METHODS

# Joint modeling of effect sizes for two correlated traits: Characterizing trait properties to enhance polygenic risk prediction

Chi Zhang[1], Geyu Zhou[1,2,3], Tianqi Chen[1], Hongyu Zhao[1,4]*

1 Department of Biostatistics, Yale University School of Public Health, New Haven, Connecticut, United States of America, 2 Department of Biological Sciences, Purdue University, West Lafayette, Indiana, United States of America, 3 Department of Statistics, Purdue University, West Lafayette, Indiana, United States of America, 4 Program of Computational Biology and Bioinformatics, Yale University, New Haven, Connecticut, United States of America

* Hongyu.zhao@yale.edu

## Abstract

Recent years have witnessed a surge in the development of innovative polygenic score (PGS) methods, driving their extensive application in disease prevention, monitoring, and treatment. However, the accuracy of genetic risk prediction remains moderate for most traits. Currently, most PGSs were built based on the summary statistics from the target trait, while many traits exhibit varied degrees of shared genetic architecture or pleiotropy. Appropriate leveraging of pleiotropy from correlated traits can potentially improve the performance of PGS of the target trait. In this study, we present PleioSDPR, a novel method that jointly models the genetic effects of complex traits and identifies conditions under which leveraging pleiotropy can improve polygenic risk prediction. PleioSDPR models the joint distribution of effect sizes across traits, allowing SNPs to be null for both traits, causal for only one trait, or causal for both traits, and it flexibly captures region-specific genetic correlations and unequal heritability across traits. Through extensive simulations and real data applications, we demonstrate that PleioSDPR improves prediction performance compared with several univariate and multivariate PGS methods, especially when there is no validation dataset. For example, by incorporating information from schizophrenia or leg fat-free mass, PleioSDPR effectively improves the prediction accuracy of bipolar disorder (14.5% accuracy gain) and hip circumference (14.6% accuracy gain), respectively. Moreover, our results indicate that traits with stronger genetic correlations to the target trait, greater heritability, and limited sample overlap contribute more substantially to enhancing prediction accuracy for the target trait. Overall, our study highlights the potential of PleioSDPR to enhance the accuracy of genetic risk prediction by effectively leveraging pleiotropy across traits and diseases. These findings contribute to a broader understanding of polygenic risk prediction and underscore the importance of incorporating pleiotropic information to improve the use of these predictions in disease prevention and treatment strategies.

**Data availability statement:** Data from the UK Biobank (www.ukbiobank.ac.uk) cannot be shared publicly, as access is restricted to approved users only. Access to data was obtained under application number 29900. GWASs for waist circumference and hip circumference (under the no sample overlap scenario), as well as WHR, T2D, BIP, and SCZ, are available online, and the papers from which they are from have been cited in the manuscript. They can be accessed through the GIANT consortium data files (https://giant-consortium.web.broadinstitute.org/index.php/GIANT_consortium_data_files), https://diagram-consortium.org/, and https://pgc.unc.edu/for-researchers/download-results/.

**Funding:** This work was supported in part by the National Institutes of Health (grant R01 HG012735 to CZ, GZ, TC, and HZ). The funders had no role in study design, data collection and analysis, decision to publish, or preparation of the manuscript.

**Competing interests:** The authors have declared that no competing interests exist.

## Author summary

Most complex human traits and diseases are influenced by many genetic variants across the genome. Although polygenic scores (PGS) have emerged as a powerful tool for predicting an individual's genetic risk, their accuracy remains limited for many traits. Much research has found that complex traits are genetically correlated, meaning they share a portion of their underlying biological basis. Existing PGS methods often use only the target trait's data and therefore miss valuable information contained in related traits.

In this study, we introduce PleioSDPR, a statistical method that improves genetic risk prediction by jointly modeling two genetically correlated traits. PleioSDPR separates genetic variants into those that affect both traits, one trait, or neither, and allows their shared genetic effects to vary across genomic regions. It also accounts for sample overlap between studies, which can otherwise bias results. Through simulations and real data analysis, we show that PleioSDPR achieves more accurate or comparable predictions than existing methods, especially when the auxiliary trait has higher heritability, stronger genetic correlation with the target trait, and includes samples not shared with the target trait. Our work demonstrates that incorporating pleiotropic information can substantially enhance the accuracy and utility of polygenic scores.

## Introduction

Polygenic score (PGS) is formulated by aggregating the estimated effects of numerous genetic markers across the human genome to assess an individual's genetic predisposition to complex traits or disorders. Recently, the significance of PGS has markedly increased as it has the potential to enhance the efficiency of population screening, refine diagnosis, and optimize disease treatment [1]. Additionally, the rapid advancement in genome-wide association studies (GWASs) has been pivotal in broadening the applications of PGS across diverse scientific fields [2,3].

However, a notable limitation of current PGS models is their primary focus on a single trait. Genetic correlations exist among various complex human traits or disorders [4–8]. This shared genetic basis, known as pleiotropy [9], inspires the generation of PGS by considering multiple traits simultaneously. Multiple multivariate PGS models have been developed and most of them were designed to improve prediction of a single trait across different populations. Although many multivariate PGS models were initially developed for cross-population prediction, their underlying modeling frameworks are not restricted to population differences. These methods jointly model multiple GWAS inputs by assuming partially shared effect-size distributions and by applying coupled shrinkage across the input datasets. When the reference panels are applied to the same population, the cross-study information sharing in these models effectively becomes cross-trait sharing. In other words, when the two input

GWAS correspond to different traits rather than different populations, the Bayesian effect-size sharing mechanism still allows each trait to borrow statistical strength from the other. This enables PRScsx and SDPRx to function as multivariate PGS approaches in a cross-trait context, leveraging pleiotropy to improve prediction performance. However, these multivariate PGS models have their limitation when applying to cross-trait context. For example, PRScsx [10] as an extension of PRScs [11] can use a continuous shrinkage prior for a couple of genetic effects across traits, but it does not consider genetic correlations. SDPRx [12] applies a Bayesian nonparametric prior to model different complex genetic architectures between two traits. However, it does not consider the heterogeneity of genetic correlation in different regions across the genome.

Although some methods, like PleioPred [13] and mtPGS [14], focusing on correlated traits have been proposed to improve PGS predictions by incorporating data from genetically correlated traits, they also have limitations. PleioPred leverages pleiotropy by assuming that the true SNP effect sizes follow a mixture distribution that accommodates four possibilities: a SNP may be causal for both traits, causal for only one trait, or non-causal for either trait. However, PleioPred assumes a simplified form of shared genetic architecture between traits and does not incorporate genetic correlations. In addition, it lacks an auto version for scenarios where a validation dataset is unavailable. mtPGS models the joint distribution of effect sizes across two traits under the assumption of a constant, genome-wide genetic correlation. However, it does not account for local genetic correlations and does not explicitly model SNPs with null effects on both traits and trait-specific null effects. Therefore, there is a need for new methods that more fully leverage information from correlated traits by incorporating local genetic heritability and local genetic correlations, while maintaining strong performance in settings without a validation dataset.

In pursuit of this goal, we introduce a novel method called PleioSDPR (https://github.com/Harmony-Chi/PleioSDPR/tree/main) that accounts for genetically correlated traits by integrating GWAS summary statistics and LD matrix with effect sizes under a hierarchical Bayesian model. PleioSDPR characterizes the joint distribution of the effect sizes of an SNP across two traits as null, trait-specific, or shared, with a correlation. It assumes that the genome can be divided into different regions, each with a different local heritability which can also be different between two traits. Additionally, it allows for different local genetic correlations across regions with shared causal effects between two traits. We compared the performance of PleioSDPR with existing methods through extensive simulations and real-data analysis, considering three levels of sample overlap. Our results show that PleioSDPR may substantially improve the prediction accuracy over existing methods for the target trait when the paired trait has a larger heritability, high genetic correlation, and a low level of sample overlap with the target trait.

## Description of method

### Overview of PleioSDPR

We consider observations of a complex trait $Y_1$ with $N_1$ samples and a complex trait $Y_2$ with $N_2$ samples, and assume that there are $N_s$ samples overlapped between the two traits. Let $X_1$ and $X_2$ be the standardized genotypes of the individuals for these two traits, so we have the following models for these two traits:

$$\begin{aligned} Y_1 &= X_1\beta_1 + \in_1, \\ Y_2 &= X_2\beta_2 + \in_2, \end{aligned} \tag{1}$$

where $\beta_1$ and $\beta_2$ are the true effect size vectors for two traits, $\in_1$ and $\in_2$ are the vectors of residuals with $\mathrm{var}(\in_1) = 1 - h_1^2$ and $\mathrm{var}(\in_2) = 1 - h_2^2$, and $h_1^2$ and $h_2^2$ are the heritability of two traits, respectively. Since the covariance of residuals can be introduced by common environmental factors because of sample overlap [6,15,16], the residual covariance can also be called environmental covariance. For non-overlapped individuals, we simply assume that their environmental covariances are zero. However, if there are sample overlaps between two traits, we need to consider environmental covariance. We

assume that the first $N_s$ individuals overlapped for these two traits and let $\rho_e$ be the environmental covariance, thus we could assume the residual covariance between these two traits as, where $i_1$ and $i_2$ are the individual indices for the first and second GWAS, respectively, assuming that the first $N_s$ samples overlap:

$$cov\left(\in_{1,i_1}, \in_{2,i_2}\right) = \begin{cases} \rho_e, & 1 \leq i_1 = i_2 \leq N_s \\ 0, & otherwise \end{cases} \tag{2}$$

The likelihood model of PleioSDPR differs from previous multivariate models that aimed to integrate data from different populations. PleioSDPR focuses on connecting marginal effect sizes in GWAS summary statistics for two traits within the same population to the true effect sizes, accounting for environmental covariance within the same population due to sample overlap. Thus, to accurately represent linkage disequilibrium (LD) and environmental covariance arising from sample overlap, we construct the following multivariate normal distribution as the likelihood function; the technical details are provided in the S1 Text.

$$\begin{pmatrix} \hat{\beta}_1|\beta_1 \\ \hat{\beta}_2|\beta_2 \end{pmatrix} \sim N\left( \begin{pmatrix} D\beta_1 \\ D\beta_2 \end{pmatrix}, \begin{pmatrix} (D+aI)/N_1 & N_s\rho_e(D+aI)/N_1N_2 \\ N_s\rho_e(D+aI)/N_1N_2 & (D+aI)/N_2 \end{pmatrix} \right), \tag{3}$$

where $\hat{\beta}_1$ and $\hat{\beta}_2$ are the marginal effect sizes, and D is the LD matrix. Same as SDPR [17] and SDPRx [12], this likelihood shrinks the off-diagonal covariance by a constant identity matrix to avoid the overestimation of effect sizes $\beta_1$ and $\beta_2$ for SNPs in high LD because of the mismatch between GWAS summary statistics and the reference panel. Following the original SDPR formula, we set $a = 0.1$ [17]. We set the overlap sample size $N_s$ as input. If the exact overlap sample size cannot be determined, we recommend using $\min(N_1^A, N_2^A)$— the minimum of the sample sizes contributed by the same cohort A to the GWASs of traits $Y_1$ and $Y_2$—as an approximation of the overlap sample size. Then, combining with the slope and the intercept from the cross-trait LDSC [6], we can estimate $\frac{N_s\rho_e}{\sqrt{N_1N_2}}$ (S1 Text).

Except for the likelihood function, our prior also characterizes the joint distribution of the effect sizes for two traits in one population.

$$\begin{pmatrix} \beta_{j1} \\ \beta_{j2} \end{pmatrix} \sim p_0 \begin{pmatrix} \delta_0 \\ \delta_0 \end{pmatrix} + p_1 \sum_{k=1}^{1000} \pi_{1k} \begin{pmatrix} N\left(0, \sigma_{1k}^2\right) \\ \delta_0 \end{pmatrix} + p_2 \sum_{k=1}^{1000} \pi_{2k} \begin{pmatrix} \delta_0 \\ N(0, \sigma_{2k}^2) \end{pmatrix} + p_3 \sum_{k=1}^{1000} \pi_{3k} N\left( \begin{pmatrix} 0 \\ 0 \end{pmatrix}, \begin{pmatrix} \sigma_{3k}^2 & r_k\sigma_{3k}\sigma_{4k} \\ r_k\sigma_{3k}\sigma_{4k} & \sigma_{4k}^2 \end{pmatrix} \right). \tag{4}$$

It assumes that an SNP for two traits can be classified into four possibilities: both null, -trait specific, or shared with correlation, with the last three parts further divided into smaller components with different local heritability and local genetic correlations. In the previous analysis exploring the genetic relationship between complex traits, we discovered variation in both local heritability and local genetic correlations across regions and traits [5]. Thus, PleioSDPR allows the variances to be different for two traits and allows heterogeneity in genetic correlations between two traits in different chromosomal regions to capture the information between complex traits better.

Following SDPRx, PleioSDPR also uses a truncated stick-breaking process [12,18] to represent the probability of assignment for the second (trait1-specific), third (trait2-specific), and fourth terms (shared with correlation), and applies truncation by setting the maximum number of mixture components to 1,000. This value serves as a computational upper bound, and in practice, only a small subset of components receives appreciable posterior mass, and increasing the truncation level of the components provides no meaningful performance gain while substantially increasing computational cost. Here we give the example of the fourth term (shared with correlation):

$$\alpha_3 \sim \text{Gamma}\,(0.1, 0.1),$$
$$V_{3k} \sim \text{Beta}\,(1, \alpha_3)\,,\ k = 1, \ldots, 1000,$$
$$\pi_{31} = V_{31},$$
$$\pi_{3k} = \prod_{m=1}^{k-1} (1 - V_{3m})\, V_{3k}, k = 2, \ldots, 1000. \tag{5}$$

We apply a Dirichlet distribution prior to the probability of each SNP belonging to each term:

$$(p_0, p_1, p_2, p_3) \sim \text{Dir}\,(1). \tag{6}$$

Priors of variance components for trait-specific terms are set to be a hierarchical inverse gamma prior:

$$\sigma_{1k}^2 \sim \text{IG}\,(0.5,\ \gamma_k)\,,$$
$$\gamma_k \sim \text{G}(0.5,\ 0.5),$$
$$\sigma_{2k}^2 \sim \text{IG}\,(0.5,\ \theta_k)\,,$$
$$\theta_k \sim \text{G}(0.5,\ 0.5). \tag{7}$$

Besides, as suggested by Zhai et al. [19], we assume the prior of variance-covariance for the fourth term as a hierarchical half-t prior:

$$\Sigma_k = \left( \left( \sigma_{3k}^2 \atop r_k \sigma_{3k} \sigma_{4k} \right) \left( r_k \sigma_{3k} \sigma_{4k} \atop \sigma_{4k}^2 \right) \right),$$
$$\Sigma_k \sim \text{IW}\,(2v + 1,\ B_k)\,,\ B_k = 4v \begin{bmatrix} \delta_k & 0 \\ 0 & \lambda_k \end{bmatrix},$$
$$\delta_k \sim \text{G}(0.5, 0.5),\ \lambda_k \sim \text{G}(0.5, 0.5). \tag{8}$$

We partitioned the LD matrix into approximately independent LD blocks to reduce the computational burden (e.g., 1,617 blocks generated with LD threshold of $r^2 > 0.1$ and the 1000 Genomes Phase 3 EUR data as reference panel) [12,20] and adopted a Markov chain Monte Carlo (MCMC) algorithm based on the Gibbs sampler (S1 Text) to estimate the posterior effect sizes with 1,000 MCMC iterations and the first 200 iterations as the burn-in.

## Simulation setting

In our simulation study, we used individual-level genotype data from unrelated white British participants in the UK Biobank (UKB) [21]. We selected SNPs that were common between the UK Biobank, 1000 Genomes Phase 3, and HapMap3 datasets, resulting in a total of M = 30,000 SNPs by selecting the first 3,000 SNPs from chromosomes 1–10. We randomly selected 20,000 individuals each to form sets 1 and 2, with no overlap between them. Then we randomly selected 10,000 individuals from set 1 and set 2 separately to construct set 3, and then used set 1 and set 3 to explore partial sample overlap scenario. We also used set 1 for both traits to represent the complete sample overlap scenario. To assess the impact of sample size on model performance, we additionally generated sets 4 and 5, each consisting of 10,000 non-overlapping individuals.

We assumed the underlying genetic architectures of the effect sizes for two traits included five parts, where 88% of the SNPs had zero effects on both traits, 2% of the SNPs had non-zero effects on only one trait, 1% of the SNPs had non-zero effects on both traits with high genetic correlation ($r^{share1} = 0.9$)and the remaining SNPs also had non-zero effects on both traits but with genetic correlation ranging from moderate to high ($r^{share2} = 0.5,\ 0.7,\ \text{and}\ 0.9$). By incorporating SNPs with different levels of correlation across specific subsets of the genome, our simulation accounted for local genetic correlations rather than assuming a constant genetic correlation across the genome.

$$\begin{pmatrix} \beta_{j1} \\ \beta_{j2} \end{pmatrix} \sim p_0 \begin{pmatrix} \delta_0 \\ \delta_0 \end{pmatrix}$$
$$+ p_1 \begin{pmatrix} N(0, \frac{h_1^2}{M(p_1+p_3+p_4)}) \\ \delta_0 \end{pmatrix}$$
$$+ p_2 \begin{pmatrix} \delta_0 \\ N(0, \frac{h_2^2}{M(p_2+p_3+p_4)}) \end{pmatrix}$$
$$+ p_3\, N\left( \begin{pmatrix} 0 \\ 0 \end{pmatrix}, \begin{pmatrix} \frac{h_1^2}{M(p_1+p_3+p_4)} & r^{shar1}\frac{h_1 h_2}{M\sqrt{(p_1+p_3+p_4)(p_2+p_3+p_4)}} \\ r^{share1}\frac{h_1 h_2}{M\sqrt{(p_1+p_3+p_4)(p_2+p_3+p_4)}} & \frac{h_2^2}{M(p_2+p_3+p_4)} \end{pmatrix} \right)$$
$$+ p_4\, N\left( \begin{pmatrix} 0 \\ 0 \end{pmatrix}, \begin{pmatrix} \frac{h_1^2}{M(p_1+p_3+p_4)} & r^{share2}\frac{h_1 h_2}{M\sqrt{(p_1+p_3+p_4)(p_2+p_3+p_4)}} \\ r^{share2}\frac{h_1 h_2}{M\sqrt{(p_1+p_3+p_4)(p_2+p_3+p_4)}} & \frac{h_2^2}{M(p_2+p_3+p_4)} \end{pmatrix} \right).$$

(9)

The heritability of the first trait was set to be $h_1^2 = 0.15$ and the heritability of the second trait was set to be $h_2^2 = 0.3$. When there were sample overlaps, we set the environmental covariance between two traits as 0.2. Then we applied GCTA [22] to generate continuous phenotypes and used PLINK 2.0 [23,24] to generate GWAS data using the training samples (set 1, set 2, set 3, set 4, or set 5). We replicated each simulation setting 10 times and evaluated the performance of different PGS methods using Pearson's correlation.

We evaluated different PGS methods across four simulation scenarios with varying levels of sample overlap and sample sizes. In the first scenario, sets 1 and 2 were used as independent training samples for two traits, ensuring no sample overlap and 20,000 individuals for each trait. In the second scenario, sets 1 and 3 were chosen as the training samples, representing a scenario with partial sample overlap, with 20,000 individuals each. In the third scenario, we selected set 1 to conduct GWASs for both traits, reflecting a scenario with complete sample overlap with 20,000 sample size. In the last scenario, sets 4 and 5 were used to construct GWASs with no sample overlap with 10,000 individuals each.

### Real GWAS data

In real data analysis, we also tried to evaluate the performance of PleioSDPR under different levels of sample overlap (No sample overlap, partial sample overlap, and complete sample overlap). We obtained GWASs from various consortia, including the Genetic Investigation of ANthropometric Traits (GIANT), DIAbetes Genetics Replication And Meta-analysis (DIAGRAM), and the Psychiatric Genomics (PGC) Consortium. To comprehensively account for different levels of sample overlap, we conducted GWASs using individual-level data from the UKB. We chose 381,924 unrelated White British individuals and selected SNPs overlapped with the Hapmap3 dataset. We conducted BOLT-LMM [25] to generate GWASs, adjusting for covariates including sex, age, age squared, sex by age interaction, sex by age squared interaction, genotyping array, and the first 20 genotype principal components.

In the scenario where there was no sample overlap between two traits, we compared PGS methods across three trait pairs — waist circumference and weight, hip circumference and leg fat-free mass, and type 2 diabetes (T2D) and waist-hip ratio (WHR) — which not only exhibit high global genetic correlations but also contain multiple regions with significant local genetic correlations of varying magnitudes. The GWASs of waist circumference, hip circumference, T2D, and WHR were obtained from publicly available datasets. We generated the GWASs of weight and leg fat-free mass using about 80% selected unrelated White British individuals in UKB. We evaluated the performance of these methods within the unrelated European population of the UK Biobank, specifically individuals born in England (identified by Field 1647 coded as 1), who were not included in the training dataset used to generate GWASs. Other unrelated European individuals in UKB that were not included in both training and testing datasets were used for parameters tuning and linear combination. To reduce potential dependence on any single validation dataset and to assess the robustness of hyperparameter tuning, we

randomly partitioned these individuals into three validation datasets of different sizes. Methods such as LDpred2, PRScs, PRScsx, and mtPGS rely on a validation dataset for tuning, and the optimal hyperparameters can vary with the size of the validation sample. Using multiple validation sets of varying sizes helps mitigate instability arising from a specific split and provides a more reliable assessment of each method's tuning performance. We validated each PGS method in different validation datasets and assessed their performance on the same testing dataset. To evaluate predictive performance, we fit a simple linear regression model of the form phenotype $\sim$ PGS in the testing dataset and reported the resulting coefficient of determination ($R^2$) which reflects the variance explained by the PGS alone. This procedure yielded three $R^2$ values—one for each validation-derived model—and their mean served as our final evaluation metric. Sample sizes for the testing and validation datasets are summarized in S1 Table.

A summary of the sample sizes, heritability, genetic correlation, and the number of regions with significant local genetic correlations for these traits is presented in Table 1. We report SNP heritability estimated from LDSC [26] for descriptive purposes. As LDSC is known to produce conservative heritability estimates and may yield different values across GWAS due to differences in sample size, LD structure, or phenotype measurement, these estimates were not used to tune any model parameters and do not affect the comparative evaluation of PGS methods.

In the scenario with partial sample overlap between two traits, we continued our comparison for waist circumference and weight, as well as hip circumference and leg fat-free mass. The GWASs of these four traits were conducted using individual-level data from the UKB. Specifically, we utilized the same GWASs for weight and leg fat-free mass as those from the real data analysis with no sample overlap. For waist circumference and hip circumference, we generated GWASs using approximately half of the individuals used for the weight and leg fat-free mass GWASs. Additionally, we introduced a new trait pair, schizophrenia (SCZ) and bipolar disorder (BIP), for which GWASs were obtained from the PGC consortium. Same as waist circumference and weight, and hip circumference and leg fat-free mass, BIP and SCZ also have several regions with significant local genetic correlations. The testing and validation datasets remained consistent with the previous real-data analysis. However, due to the limited number of cases of BIP and SCZ in the UKB, we considered only two validation datasets for this trait pair. Sample sizes for the testing and validation datasets are summarized in S2 Table. A summary of the sample sizes of each trait, overlapping sample size, heritability, global genetic correlation, and the number of regions with significant local genetic correlations for these traits is provided in Table 2.

When the samples for two traits completely overlapped, we only compared waist circumference and weight, as well as hip circumference and leg fat-free mass. The GWASs of these four traits were conducted using individual-level data from the UKB with the same sample sizes. The validation and testing datasets remained the same as the previous real data analysis, and sample sizes for the testing and validation datasets are summarized in S3 Table. A summary of the sample sizes of each trait, overlapping sample size, heritability, global genetic correlation and the number of regions with significant local genetic correlations for these traits is provided in Table 3.

**Table 1. Trait-pairs for real data analysis without sample overlap.** We provide the sample sizes, the heritability, and the source of GWASs for these six traits. We also show the genetic correlation between each trait pair. Heritability and genetic correlation were calculated from LDSC [6,26]. The number of regions with significant local genetic correlations (p-value< 0.05/ number of regions) was inferred from SUPERGNOVA [4].

| Trait 1 | Sample size for trait 1 | Heritability for trait 1 (s.e.) | Trait 2 | Sample size for trait 2 | Heritability for trait 2 (s.e.) | global genetic correlation between two traits (s.e.) | number of regions with significant local genetic correlations |
|---|---|---|---|---|---|---|---|
| Waist circumference (GIANT) [27] | 232,101 | 0.14 (0.009) | Weight (UKB) | 304,503 | 0.28 (0.011) | 0.90 (0.013) | 47 |
| Hip circumference (GIANT) [27] | 213,038 | 0.15 (0.009) | Leg fat-free mass (UKB) | 299,986 | 0.28 (0.012) | 0.87 (0.015) | 36 |
| Type-2 diabetes (DIAGRAM) [28] | 305,404 | 0.08 (0.006) | Waist-hip ratio (GIANT) [27] | 304,809 | 0.10 (0.006) | 0.54 (0.034) | 2 |

**Table 2. Trait-pairs for real data analysis with partial sample overlap.** We provide the sample sizes of each trait, the overlapped sample sizes, the heritability, and the source of GWASs for these six traits. We also show the genetic correlation between each trait pair. Heritability and genetic correlation were calculated from LDSC [6,26]. The number of regions with significant local genetic correlations (p-value< 0.05/ number of regions) was inferred from SUPERGNOVA [4].

| Trait 1 | Sample size for trait 1 | Heritability for trait 1 (s.e.) | Trait 2 | Sample size for trait 2 | Heritability for trait 2 (s.e.) | Overlap sample size | genetic correlation between two traits (s.e.) | number of regions with significant local genetic correlations |
|---|---|---|---|---|---|---|---|---|
| Waist circumference (UKB) | 152,117 | 0.21 (0.009) | Weight (UKB) | 304,503 | 0.28 (0.010) | 151,900 | 0.90 (0.008) | 27 |
| Hip circumference (UKB) | 152,384 | 0.21 (0.009) | Leg fat-free mass (UKB) | 299,986 | 0.28 (0.011) | 149,711 | 0.85 (0.011) | 31 |
| Bipolar Disorder (PGC) [29] | 353,899 | 0.08 (0.003) | Schizophrenia (PGC) [30] | 130,644 | 0.37 (0.013) | 130,644 | 0.70 (0.015) | 27 |

**Table 3. Trait-pairs for real data analysis with complete sample overlap.** We provide the sample sizes of each trait, the overlapped sample size, the heritability, and the source of GWASs for these six traits. We also show the genetic correlation between each trait pair. Heritability and genetic correlation were calculated from LDSC [6,26]. The number of regions with significant local genetic correlation (p-value< 0.05/ number of regions) was inferred from SUPERGNOVA [4].

| Trait 1 | Sample size for trait 1 | Heritability for trait 1 (s.e.) | Trait 2 | Sample size for trait 2 | Heritability for trait 2 (s.e.) | Overlap sample size | genetic correlation between two traits (s.e.) | number of regions with significant local genetic correlations |
|---|---|---|---|---|---|---|---|---|
| Waist circumference (UKB) | 304,426 | 0.21 (0.008) | Weight (UKB) | 304,426 | 0.28 (0.010) | 304,426 | 0.91 (0.005) | 30 |
| Hip circumference (UKB) | 299,943 | 0.22 (0.009) | Leg fat-free mass (UKB) | 299,943 | 0.28 (0.011) | 299,943 | 0.84 (0.008) | 38 |

Tables 1–3 show that the selected trait pairs exhibit unbalanced heritability and high global genetic correlations. Moreover, results from SUPERGNOVA indicate that most pairs display both significant and heterogeneous local genetic correlations. We assessed this heterogeneity by examining the distribution of significant local genetic correlations across genomic regions. As shown in S1 Fig, these local estimates span a wide range across regions, demonstrating heterogeneity in regional genetic sharing rather than a single uniform correlation structure.

## Verification and comparison

### Compared methods

In our analysis, we evaluated the performance of PleioSDPR with seven other approaches: MTAG [31], shaPRS [32], LDpred2 [33], PRScs [11], PRScsx [10], SDPRx [12], and mtPGS [14]. These methods fall into three primary categories. The first category combines multi-trait GWAS analysis (e.g., MTAG or shaPRS) with subsequent application of a PGS method to the resulting summary statistics. For downstream PGS construction, we use PRScs, as it is widely used and well-established. The second category includes univariate PGS models (LDpred2, PRScs) that use GWAS data from a trait as input. The third category includes multivariate PGS methods (PRScsx, SDPRx, and mtPGS). We selected these methods for comparison because all of them can be applied directly to GWAS summary statistics, are developed for two GWASs as input, and, importantly, each of them includes an implementation that does not require a validation dataset. When comparing the performance of different methods, we considered two distinct scenarios: one that incorporated a validation dataset and another that did not. Without a validation set, the strategy of the linear combination of PGSs from multiple traits and tuning the methods' parameters is infeasible. Thus, when no validation dataset was available, we applied the auto-tuning versions of the downstream PRScs results for MTAG and shaPRS (MTAG-single, shaPRS-single). For LDpred2, PRScs, and PRScsx, we used their respective auto versions (LDpred2-single, PRScs-single, PRScsx-single).

SDPRx and PleioSDPR do not require parameter tuning, so we directly applied their default settings (SDPRx-single, PleioSDPR-single). For mtPGS, we followed the recommendations in the original paper and set the hyperparameters to $p = 1 \times 10^{-6}$ and $r^2 = 0.2$ (mtPGS-single).

When there was a validation dataset, for all the methods a linear combination of PGSs from different traits was performed to derive the final score for the target trait (MTAG-mult, shaPRS-mult, Ldpred2-mult, PRScs-mult, PRScsx-mult, mtPGS-mult, SDPRx-mult, and PleioSDPR-mult). For PRScsx and PRScs, the global shrinkage parameter was specified as {1e−06, 1e−04, 1e−02, 1, auto}. For LDpred2, we ran LDpred2-inf, LDpred2-auto, and LDpred2-grid and reported the best performance of the three options. The grid of hyperparameters of LDpred2-grid was set as non-sparse, $p$ in a sequence of 17 values from 1e-05–1 on a log scale, and $h^2$ within {0.7, 1, 1.4} of $h^2_{LDSC}$. In mtPGS, we tune hyperparameters, $p$ and $r^2$, in the validation set based on four different values of $p$ as {1e−5, 1e−6, 1e−7, and 1e−8} and three different values of $r^2$ as {0.1, 0.2, and 0.25}.

## Simulation results

We began by evaluating the predictive performance of each method through simulations across multiple genetic correlations and sample overlap levels. We considered eight methods: PleioSDPR, SDPRx, mtPGS, PRScsx, PRScs, LDpred2, shaPRS, and MTAG. shaPRS and MTAG are meta-GWAS methods that integrate GWASs of two traits and then apply PRScs to the resulting GWASs for PGS analysis. LDpred2 and PRScs are univariate methods that utilize summary statistics from a single trait, while PleioSDPR, SDPRx, mtPGS, and PRScsx are multivariate methods that jointly incorporate GWAS summary statistics from multiple traits. For simulation studies, we used genotype data from unrelated White British individuals from the UKB. We selected a total of 30,000 SNPs, with 3,000 SNPs from each of chromosomes 1–10, respectively. The training cohort comprised 20,000 individuals for both traits. We considered three levels of sample overlap: no overlap, partial overlap (with an overlapped sample size of 10,000), and complete overlap. We also consider the training cohort comprised 10,000 individuals for both traits without sample overlap. The validation and test datasets each consisted of 10,000 individuals. The genetic architecture simulated for the two traits was detailed in the Description of method section, including unbalanced heritability (0.15 and 0.3) and different values of genetic correlations. Each simulation setting was repeated 10 times. We first generated summary statistics for two traits without sample overlap and used the EUR 1 KG phase 3 data [34] as the reference panel to estimate the LD matrix. We then evaluated the performance of each method using the square of the Pearson correlation of PGS and simulated phenotype in the independent testing dataset. In cases where a separate validation dataset was available, we used it to tune the parameters for LDpred2, PRScs, mtPGS, and PRScsx. We also performed linear combinations of the PGSs of the two traits. When no validation dataset was available, we used the auto versions of LDpred2, PRScs, and PRScsx, the version of mtPGS with $p = 1 \times 10^{-6}$ and $r^2 = 0.2$, as well as the original versions of SDPRx and PleioSDPR, since parameter tuning was unnecessary for SDPRx and PleioSDPR.

Fig 1 presents the outcomes of simulations conducted for scenarios where two traits do not share samples with 20,000 sample size for each trait. We first compared the performance of PGS methods without a validation dataset (with the "_single" suffix). PleioSDPR consistently outperformed the other PGS methods across all genetic correlations, particularly for traits with lower heritability (S4 Table). The results also reveal that the greater the genetic correlation between the two traits, the more accurate PleioSDPR predictions are. We also observed that PleioSDPR, which modeled local genetic correlation and flexible trait-specific variances, outperformed SDPRx which assumed a constant genetic correlation and equal variances for two traits, mtPGS which assumed a constant genetic correlation and PRScsx which did not account for genetic correlation. This finding underscores the importance of modeling local genetic correlations, as accounting for region-specific genetic architectures enables more accurate identification of trait-specific and shared genetic effects, ultimately leading to improved polygenic score prediction. We further observed that MTAG and shaPRS outperformed PRScs alone, supporting the idea that integrating GWAS data prior to applying a PGS method can improve prediction performance.

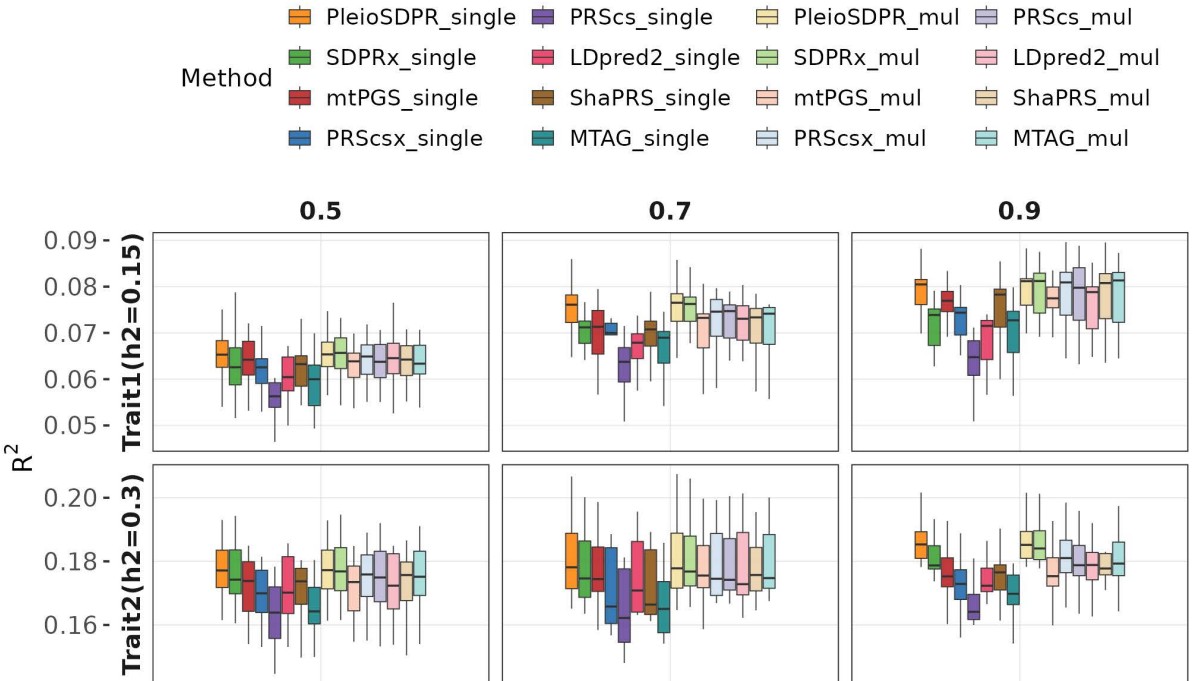

**Fig 1. Simulation results when there is no sample overlap with 20,000 sample sizes for both traits.** We compare the methods considering with the validation dataset and without the validation dataset. When there is no validation dataset, we name them as PleioSDPR_single, SDPRx_single, mtPGS_single PRScsx_single, PRScs_single, LDpred2_single, shaPRS_single and MTAG_single. When there is a validation dataset, we name them as PleioSDPR_mul, SDPRx_mul, mtPGS_mul, PRScsx_mul, PRScs_mul, LDpred2_mul, shaPRS_mul and MTAG_mul. Exact $R^2$ values for all methods are provided in S4 Table.

For methods with "_mul" suffix, which means the inclusion of a validation dataset, there is a notable increase in accuracy for all methods except PleioSDPR (S4 Table). This suggests that these methods, except PleioSDPR, benefit significantly from the validation dataset. In contrast, PleioSDPR maintains a relatively stable accuracy no matter whether the validation dataset exists or not, and consistently performs the best among all the compared methods. This indicates that PleioSDPR can learn sufficient information directly from GWAS data, making the validation dataset less necessary for improving performance compared to the other methods. When analyzing the simulation results under conditions of partial sample overlap (S2 Fig and S5 Table), we found that PleioSDPR maintains its superiority over alternative methods in scenarios without a validation dataset. However, this advantage is less obvious in cases with complete sample overlap between two traits (S3 Fig and S6 Table). With the introduction of a validation dataset, different methods yielded similar performance, with PleioSDPR consistently achieving or matching the highest levels of predictive accuracy.

When the samples of two traits completely overlapped, PleioSDPR performed exclusively one of the best when the genetic correlation of two traits was high (0.9), when the prediction was on the first trait with lower heritability in the absence of the validation dataset (S6 Table). Given that the two traits fully overlap, information from correlated traits due to increased sample size is not possible; instead, the method benefits from information from the traits with higher heritability. This suggests that to demonstrate optimal performance of PleioSDPR, there must be sufficient information from the paired correlated traits to be borrowed to improve prediction of the target trait. In simulations of complete sample overlap, PleioSDPR does not outperform all competing methods; however, it at least shows performance comparable with the other methods (S3 Fig and S6 Table).

To evaluate the influence of GWAS sample size on model performance, we conducted an additional simulation in which each trait had a sample size of 10,000 (S4 Fig and S7 Table). In this smaller-sample scenario, estimation of local genetic correlations and heritability becomes less stable, leading to modest attenuation in the performance of models that rely on these quantities, including PleioSDPR. Even so, PleioSDPR continues to perform comparably to other multivariate PGS models across all genetic-correlation settings, and mtPGS shows stronger performance when sample sizes are small. Overall, these results indicate that PleioSDPR is generally robust to moderate reductions in sample size, although larger sample sizes improve the stability of correlation estimation and enhance predictive performance.

## Applications

Subsequently, we evaluated PleioSDPR's performance through real data analysis across various trait pairs. We examined the effect of varying degrees of sample overlap and evaluated the performance of these methods both with and without validation data. We used validation datasets of varying sample sizes to tune parameters and evaluated performance on the same test dataset. We used the mean of $R^2$ to quantify prediction accuracy.

We first selected three pairs of traits with no overlapping samples: Waist Circumference and Weight, Hip Circumference and Leg Fat-Free Mass, and T2D and WHR. PleioSDPR demonstrates the best performance across all six traits (Fig 2). Within these trait pairs, waist circumference, hip circumference, and T2D exhibit lower heritability compared to their auxiliary traits, and PleioSDPR demonstrated more pronounced improvement for these lower-heritability traits than other multivariate PGS methods. PleioSDPR outperformed SDPRx, mtPGS and PRScsx, which did not account for local genetic correlations. For all three trait pairs, local genetic correlation analyses revealed multiple regions with significant and varying local genetic correlation values. These findings emphasize that incorporating heterogeneous local genetic correlations into modeling can improve polygenic score prediction, particularly for traits with lower heritability. When a validation dataset was used for parameter tuning (S5 Fig), PleioSDPR still performed best or second-best with comparable performance to other PGS methods.

Environmental covariance is important when samples overlap between pairs of traits. To investigate the impact of sample overlap, we generated GWAS summary statistics in UKB for Waist Circumference and Weight, Hip Circumference,

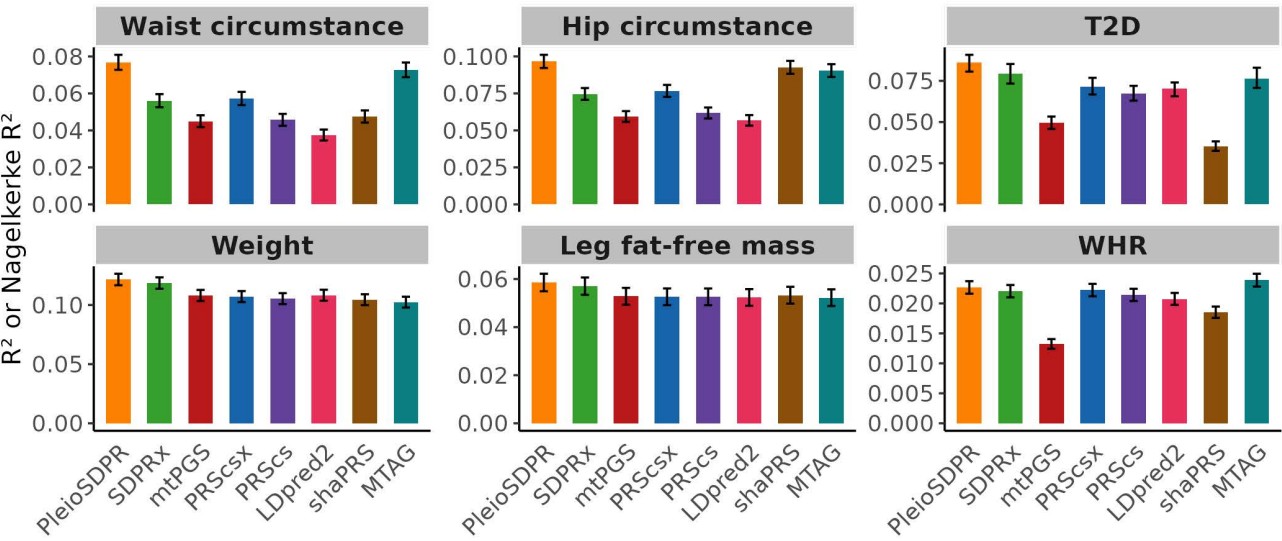

**Fig 2. Prediction performance of different methods for three trait pairs that have no sample overlap when there is no validation dataset.** Error bars represent 95% confidence intervals.

and Leg Fat-Free Mass. Additionally, BIP and SCZ were also considered under the condition of partial sample overlap. We first evaluated the performance of PleioSDPR with and without the accounting for environmental covariance. When incorporating environmental covariance, we estimate $s = \frac{N_s \rho_e}{\sqrt{N_1 N_2}}$ as described in the S1 Text. When environmental covariance is not considered, we set $s = 0$, i.e., assume $N_s = 0$. The comparison indicates that the inclusion of the estimation of environmental covariance enhances PleioSDPR's performance (S6 and S8 Figs). We next compared the performance of PleioSDPR with other methods when there is sample overlap. In the absence of a validation dataset, PleioSDPR exhibited the best performance in predicting Waist Circumference, Hip Circumference, BIP, and SCZ while ranking second in accuracy for Weight and Leg Fat-Free Mass (Fig 3). This pattern mirrors the observations made in the simulation scenario where PleioSDPR demonstrates a predictive advantage with traits that have lower heritability.

When incorporating a validation dataset (S7 Fig), PleioSDPR remained the top-performing method for Waist Circumference, Hip Circumference, BIP, and SCZ, and ranked as the second most accurate for the remaining traits. However, across all traits, PleioSDPR's performance was comparable with SDPRx when the validation dataset is considered.

When the samples for two traits were entirely overlapping, we focused on two specific pairs: Waist Circumference and Weight, and Hip Circumference and Leg Fat-Free Mass. To ensure complete sample overlap across these trait pairs, we used the same UKB cohort to generate GWASs for these traits. Our findings indicate that the performance of all methods is comparable across all traits, regardless of whether a validation dataset is considered. In this context, no single method distinctly outperformed the others (S9 and S10 Figs).

## Discussion

We have introduced a novel PGS model, PleioSDPR, designed to improve the prediction accuracy of the target trait by leveraging genetically correlated traits. PleioSDPR offers flexibility in modeling the genetic architecture between two complex traits, accounting for variations in local genetic heritability and local genetic correlations across different regions. Moreover, PleioSDPR considers the environmental covariance introduced by sample overlap by modifying the likelihood function that can be used in the MCMC algorithm. Notably, PleioSDPR eliminates the need for parameter tuning and maintains optimal performance when there is no validation dataset.

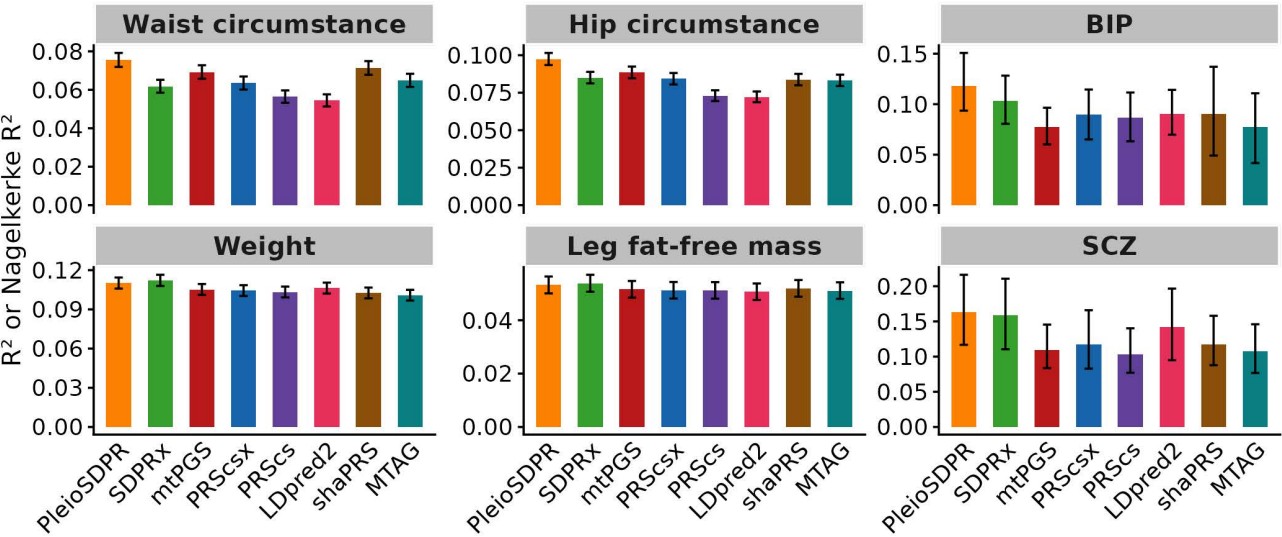

**Fig 3. Prediction performance of different methods for three trait pairs that have partial sample overlap when there is no validation dataset.** Error bars represent 95% confidence intervals.

Building on the Bayesian nonparametric prior concept from SDPRx, PleioSDPR further explores the shared genetic architecture between the target trait and correlated traits by classifying SNPs into different categories according to their effect sizes. PleioSDPR assumes that SNPs may either be non-causal for both traits, causal for only one, or causal for both. When SNPs are categorized as causal for at least one trait, they are further divided into subcategories with different variances or covariances. Besides, by addressing the limitations of SDPRx, PleioSDPR allows for variation in the variance of SNPs that are shared between traits and accommodates different local genetic correlations within different subcategories. This method of SNP categorization enables PleioSDPR to apply shrinkage to the estimated effect sizes across different SNP categories, ensuring adequate shrinkage for small effects while preventing over-shrinkage of large effects, particularly in SNPs that are causal for both traits.

In this study, we evaluated the performance of PleioSDPR through both simulations and real data analyses. We found that selecting GWAS data from correlated traits that not only share similarities with the target trait but also provide complementary genetic information is critical for improving prediction accuracy. Specifically, appropriately leveraging pleiotropy—by utilizing shared genetic effects, regions of overlapping causality, and supplementary heritability information—can substantially enhance the performance of polygenic risk scores, particularly when the auxiliary trait exhibits high genetic correlation, substantial heritability, and minimal sample overlap with the target trait. Our simulation settings and real data applications considered scenarios involving two correlated traits with unbalanced heritability, moderate to high genetic correlation, varying degrees of sample overlap and multiple regions with different significant local genetic correlations. These results highlight that incorporating pleiotropic information can meaningfully improve genetic risk prediction for complex traits. In particular, modeling heterogeneous genetic correlations across genomic regions better captures the underlying genetic architecture, where local correlation patterns vary across the genome. However, introducing region-specific parameters also increases estimation variance. Without sufficient information borrowing across traits and regions, the additional variance from modeling multiple local correlations may offset the benefits, potentially resulting in comparable or even diminished performance relative to simpler models such as SDPRx. Furthermore, explicitly accounting for environmental covariance was shown to further enhance prediction precision in the presence of overlapping samples. Overall, our findings demonstrate that PleioSDPR is particularly effective when the auxiliary trait has high genetic correlation, greater heritability, and minimal sample overlap with the target trait.

Beyond summary-statistic–based models, several recently proposed frameworks—such as multi-PGS [35] and PRSmix+ [36] also aim to leverage correlated traits to improve prediction accuracy. These methods construct large libraries of single-trait PGSs and require a validation cohort to learn optimal trait-specific weights. While highly effective in data-rich biobank settings, these approaches do not operate directly on GWAS summary statistics. PleioSDPR, in contrast, is designed for scenarios where only summary-level information is available and focuses specifically on modeling two traits through shared and region-specific genetic architecture without using a validation dataset. Thus, multi-PGS and PRSmix + are complementary rather than directly comparable to PleioSDPR. JointPRS [37], which extends PRScsx [10], is another multivariate PGS framework that leverages genetic correlations across multiple populations and can, in principle, incorporate more than two GWAS datasets. Although JointPRS is related in spirit to multivariate PGS modeling, it is primarily designed for settings involving multiple GWAS inputs, whereas the present study focuses specifically on the two-trait cross-trait scenario. Therefore, JointPRS may become a relevant comparison in future extensions of our method that accommodate more than two traits.

However, we have identified several limitations that will be addressed in future research. First, PleioSDPR currently requires estimating a multitude of parameters, which may increase variance and instability in the model. Future work will involve pinpointing and simplifying components of the model that are unnecessarily complex, potentially reducing the number of parameters needed. Additionally, the PleioSDPR computational process is rather time-consuming, even when leveraging parallel computation across 22 chromosomes with the allocation of six threads per chromosome. We

have quantified this limitation more explicitly in S8 Table, which compares the runtime of PleioSDPR with other competing methods under a representative simulation setting. To enhance efficiency, we are exploring the possibility of rewriting the code using C++ programming. Lastly, PleioSDPR is limited to analyzing only two correlated traits simultaneously. Moving forward, we will extend the model's capability to include multiple correlated traits, broadening the scope of our genetic architecture analysis.

## Supporting information

**S1 Text. Supplementary method.**
(DOCX)

**S1 Fig. Distribution of significant local genetic correlations.** Local genetic correlations are estimated by SUPERGNOVA and the significant local genetic correlations are selected with pvalue < 0.05/number of regions used for each trait pair. The above distribution of local genetic correlations integrates results from all trait-pairs used in the real application.
(TIF)

**S2 Fig. Simulation results when there is partial sample overlap with 20,000 sample sizes for each trait.** We compare the methods considering with validation dataset and without validation dataset. When there is no validation dataset, we name them as PleioSDPR_single, SDPRx_single, mtPGS_single, PRScsx_single, PRScs_single, LDpred2_single, ShaPRS_single and MTAG_single. When there is a validation dataset, we name them PleioSDPR_mul, SDPRx_mul, mtPGS_mul, PRScsx_mul, PRScs_mul, LDpred2_mul, ShaPRS_mul and MTAG_mul. Exact $R^2$ values for all methods are provided in S5 Table.
(TIF)

**S3 Fig. Simulation results when there is complete sample overlap with 20,000 sample sizes for each trait.** We compare the methods considering with validation dataset and without validation dataset. When there is no validation dataset, we name them as PleioSDPR_single, SDPRx_single, mtPGS_single, PRScsx_single, PRScs_single, LDpred2_single, ShaPRS_single and MTAG_single. When there is a validation dataset, we name them PleioSDPR_mul, SDPRx_mul, mtPGS_mul, PRScsx_mul, PRScs_mul, LDpred2_mul, ShaPRS_mul and MTAG_mul. Exact $R^2$ values for all methods are provided in S6 Table.
(TIF)

**S4 Fig. Simulation results when there is no sample overlap with 10,000 sample sizes for each trait.** We compare the methods considering with validation dataset and without validation dataset. When there is no validation dataset, we name them as PleioSDPR_single, SDPRx_single, mtPGS_single, PRScsx_single, PRScs_single, LDpred2_single, ShaPRS_single and MTAG_single. When there is a validation dataset, we name them PleioSDPR_mul, SDPRx_mul, mtPGS_mul, PRScsx_mul, PRScs_mul, LDpred2_mul, ShaPRS_mul and MTAG_mul. Exact $R^2$ values for all methods are provided in S7 Table.
(TIF)

**S5 Fig. Prediction performance of different methods for 3 trait pairs that have no sample overlap when there is a validation dataset.** Error bars represent 95% confidence intervals.
(TIF)

**S6 Fig. The comparison of PleioSDPR before and after considering sample overlap in its model when two traits have partial sample overlap.** Error bars represent 95% confidence intervals.
(TIF)

**S7 Fig. Prediction performance of different methods for 3 trait pairs that have partial sample overlap when there is a validation dataset.** Error bars represent 95% confidence intervals.
(TIF)

**S8 Fig. The comparison of PleioSDPR before and after considering sample overlap in its model when two traits have complete sample overlap.** Error bars represent 95% confidence intervals.
(TIF)

**S9 Fig. Prediction performance of different methods for 2 trait pairs that have complete sample overlap when there is no validation dataset.** Error bars represent 95% confidence intervals.
(TIF)

**S10 Fig. Prediction performance of different methods for 2 trait pairs that have complete sample overlap when there is a validation dataset.** Error bars represent 95% confidence intervals.
(TIF)

**S1 Table. Sample sizes of testing and validation datasets when two traits in a trait pair have no sample overlaps.**
(XLSX)

**S2 Table. Sample sizes of testing and validation datasets when two traits in a trait pair have partial samples overlapped.**
(XLSX)

**S3 Table. Sample sizes of testing and validation datasets when two traits in a trait pair have complete samples overlapped.**
(XLSX)

**S4 Table. Simulation results when there is no sample overlap with 20,000 sample sizes for both traits.**
(XLSX)

**S5 Table. Simulation results when there is partial sample overlap with 20,000 sample sizes for both traits.**
(XLSX)

**S6 Table. Simulation results when there is complete sample overlap with 20,000 sample sizes for both traits.**
(XLSX)

**S7 Table. Simulation results when there is no sample overlap with 10,000 sample sizes for both traits.**
(XLSX)

**S8 Table. Runtime of each method in one simulation.**
(XLSX)

## Acknowledgments

We conducted the research by using the UK Biobank resource under an approved data request (ref: 29900). We sincerely thank the GIANT, the DIAGRAM consortia and the PGC consortium for making their GWAS summary data publicly accessible. We thank Leqi Xu for her insightful suggestions.

## Author contributions

**Conceptualization:** Chi Zhang, Geyu Zhou, Hongyu Zhao.

**Data curation:** Chi Zhang.

**Formal analysis:** Chi Zhang.

**Funding acquisition:** Hongyu Zhao.

**Investigation:** Chi Zhang.

**Methodology:** Chi Zhang, Geyu Zhou, Tianqi Chen, Hongyu Zhao.

**Project administration:** Hongyu Zhao.

**Resources:** Hongyu Zhao.

**Software:** Chi Zhang, Geyu Zhou.

**Supervision:** Hongyu Zhao.

**Validation:** Chi Zhang.

**Visualization:** Chi Zhang.

**Writing – original draft:** Chi Zhang.

**Writing – review & editing:** Chi Zhang, Geyu Zhou, Tianqi Chen, Hongyu Zhao.

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
