## [Decision Letter · Decision Letter 0]

25 Jun 2025

PGENETICS-D-25-00565

Joint Modeling of Effect Sizes for Two Correlated Traits Characterizing Trait Properties to Enhance Polygenic Risk

PLOS Genetics

Dear Dr. Zhao,

Thank you for submitting your manuscript to PLOS Genetics. After careful consideration, we feel that it has merit but does not fully meet PLOS Genetics's publication criteria as it currently stands. Therefore, we invite you to submit a revised version of the manuscript that addresses the points raised during the review process.

Please submit your revised manuscript within 30 days Jul 25 2025 11:59PM. If you will need more time than this to complete your revisions, please reply to this message or contact the journal office at plosgenetics@plos.org. Please include the following items when submitting your revised manuscript:

We look forward to receiving your revised manuscript.

Kind regards,

Chaolong Wang

Academic Editor

PLOS Genetics

Xiaofeng Zhu

Section Editor

PLOS Genetics

Aimée Dudley

Editor-in-Chief

PLOS Genetics

Anne Goriely

Editor-in-Chief

PLOS Genetics

**Journal Requirements:**

2) Please ensure that the Title in your manuscript file and the Title provided in your online submission form are the same.

https://journals.plos.org/plosgenetics/s/submission-guidelines#loc-parts-of-a-submission

4) Your manuscript is missing the following sections: Verification and Comparison, Applications, and Acknowledgements. Please ensure that your article adheres to the standard Methods article layout and order of Abstract, Author Summary, Introduction, Description of the Method, Verification and Comparison, Applications, Discussion, Acknowledgements, References, and Supplementary Information. For details on what each section should contain, see our Methods article guidelines:

https://journals.plos.org/plosgenetics/s/submission-guidelines#loc-manuscript-organization.

5) Please upload all main figures as separate Figure files in .tif or .eps format. For more information about how to convert and format your figure files please see our guidelines: 

6) We have noticed that you have uploaded Supporting Information files, but you have not included a list of legends. Please add a full list of legends for your Supporting Information files after the references list.

7) Thank you for providing your Data Availability Statement. We noted that this link "https://portals.broadinstitute.org/collaboration/giant/index.php/GIANT_consortium_data

_files" reaches a Not Found page. Please amend this to a working link.

8) Please amend your detailed Financial Disclosure statement. This is published with the article. It must therefore be completed in full sentences and contain the exact wording you wish to be published.

**Reviewers' comments:**

Reviewer's Responses to Questions

Reviewer #1: Comments:

The authors present a novel cross-traits polygenic score (PGS), PleioSDPR, to leverage pleiotropy from correlated traits to enhance the prediction performance of target trait. It allows for different local genetic correlations for regions having causal effects shared between two traits under a hierarchical Bayesian model. Among simulations and real-data analysis, PleioSDPR may substantially improve the prediction accuracy over existing methods for the target trait. I have some comments as follows:

Major Comments

Page 11, line 145: In the part of “Compared Methods”, the authors applied PRS-CSx and SDPRX to represent multivariate PGS method. However, both of them are designed for cross-ancestries PGS model but not cross-traits. The famous and representative cross-traits PGS category did not be considered, like MTAG (2017, NG) or shaPRS (2024, AJHG), which firstly integrating GWAS summary statistics of two traits and then apply single-trait PGS method. Could the authors add these methods for fair comparison?

Page12, line 167. In the part of “Simulations”, the authors fixed the GWAS sample size to be 20,000 among all simulations without exploring other sample size scenarios. The GWAS sample size may affects the robustness of local (and global) genetic correlation between two traits, which may enhance the performance of your method. Could the authors add the corresponding scenarios or explain why alternative GWAS sample size scenarios were not explored?

Page13, line 183. With r2 = 0.9, this simulation scenario likely reflects constant genetic correlations across the genome, which aligns with the underlying assumptions of SDPRX (Page12, line 152). However, in the constant genetic correlations setting (the third column in Figure 1), PleioSDPR_single SDPRX_single should have shown similar performance but PleioSDPR_single outperformed SDPRX_single. Could you explain such difference?

Page 15, line 224. The number of regions with significant local genetic correlation for trait pairs were calculated from SUPERGNOVA. It seems that the P < 0.05 only means this region have significant local genetic correlation but not emphasis the heterogeneous local genetic correlations. Could you explain how the heterogeneity local genetic correlations are specifically manifested?

Page 26, line 439. Apart from the prediction performance, the comparison of computation efficiency between methods is still important. Please add the corresponding materials to enhance clarity for our readers.

Minor Comments

Page13, line 179. The expression of genetic correlation r1 appears to conflict with the rk in formula (4) in Page 9, line 119. Please verify these expressions.

Page 21, Line 331. It is hard to observe the difference between methods, especially for those “mul” methods in Figure 1 (also for supplement Figure 1 and 2). Please consider adding the R² values to Figures for better clarity.

Page 23, Line 364. “…without the consideration of environmental covariance” means simply rho(e) = 0?

Supplement Figures and Tables. The title of Supplement Figure 6 should be corrected as “…have complete sample overlap” but not “…have partial sample overlap”.

Supplement Method. Page 13, Line 1. The formula of rho(t) should be corrected as rho(t) = rho(g) + rho(e).

Reviewer #2: Zhang and colleagues present pleioSDPR, a novel method for polygenic scoring which leverages pleiotropic information from correlated traits to improve prediction. They model posterior SNP effects as a four-component mixture, incorporating local heritability and genetic correlation while also correcting for sample overlap. Overall, this work is an important contribution to the literature, finding improvement over existing methods for both continuous and binary traits using real data, especially when genetic correlation is high, heritability of the donor trait is high, and there is little sample overlap. I have several questions and suggestions for the authors to consider, roughly in order of their appearance in the text.

1) Line 53: does SDPRX model cross-population or cross-trait effects? My understanding was that it is for cross-population.

2) Line 55: I assume this should be SDPRX rather than PRS-CSx?

3) There are several recent methods which leverage multiple cross-trait PGS to improve prediction, e.g. multi-PGS (Albinana et al., 2023 Nat. Comms.) and PRSmix+ (Truong et al., 2024, Cell Genomics). Although these methods work on matrices of PGS rather than starting with summary statistics and also require a validation set, they have the same goal of leveraging correlated traits to improve PGS prediction. How do these methods relate to your current work and is it appropriate to consider them as competing approaches?

4) Line 101: a closing bracket appears to be missing. Also, what does j refer to here? Later it indexes SNPs, but it doesn’t seem to do that here.

5) Line 113: you choose to set a to 0.1. Is there a strong justification for choosing this value? How would other values affect the results and how should one choose this value?

6) How is PRS-CSx used in this work? It appears that only EUR samples were used for the GWAS, wheras PRS-CSx is traditionally used with sumstats from multiple ancestries. Were the two traits treated as if they were from different ancestries, while using the same reference panel?

7) Similarly, JointPRS is mentioned in the introduction, but not used as a comparison method. If multiple ancestry sumstats are indeed being used, this would be odd to leave out after mentioning it in the intro.

8) It seems unnecessary to use 3 differently sized validation sets if the R2 values are ultimately averaged. Is there a reason for this?

9) Line 340: The calculation of R2 deserves a bit more explanation. e.g. Do you control for PCs? Is this the R2 attributable to the PRS alone or to the full model with covariates?

10) For real data figures, confidence intervals should be included around point estimates for R2.

Minor points:

1) Line 26: Usually, the term bipolar “disorder” is preferred rather than “disease”.

2) Line 316: Should this read “complete” instead of “no” sample overlap?

Reviewer #3: Recent advances in polygenic score (PGS) methods have expanded their use in disease prevention and treatment. However, prediction accuracy remains moderate for most traits. By leveraging the pleiotropy between traits with share genetic architecture can improve PGS performance. The authors proposed a new method, PleioSDPR, which jointly models genetic effects across traits and shows improved prediction accuracy over existing approaches, especially without validation datasets. Results demonstrate notable gains in PGS prediction accuracy for examples of paired traits, emphasizing the value of using pleiotropic information in genetic risk prediction. Overall, the manuscript was well written, the simulation and real data applications support the main conclusions. I have a few comments and suggestions for the authors to consider.

• Introduction: SDPRX is introduced with several PGS methods focusing on a single trait. It fits better with PleioPred and mtPGS which construct PGS considering correlated traits.

• Introduction: line 63 “follow a four-component normal distribution” it is helpful to briefly describe these four components to understand the method’s unique feature.

• Methods: line 115-116 “the minimal sample sizes of the individuals from the same cohort of two traits as the input of overlap sample size” the authors need to clarify what “minimal sample sizes” means, probably with some notations for two traits (e.g., Ny1 and Ny2 indicate the sample size of trait y1 and y2, respectively).

• Methods: for equation (4), it would be helpful to justify why k is capped at 1,000. Recent large GWAS have reported a large number of significant loci > 1,000. What would be the impact if k >1,000? If it is an arbitrary choice of computational parameters, maybe ranking these significant genetic associations could minimize the overall impact of the performance.

• Methods: line 140-141 “We partitioned the LD matrix into approximately independent LD blocks to reduce the computational burden”. How many LD blocks were calculated? The number is dependent on the LD threshold and ancestry populations.

• PGS method comparison: PleioPred and mtPGS were two very relevant methods which consider correlated traits, similar to PleioSDPR. However, the authors chose not to include them for comparison. The authors need to include them for comparison, or clearly justify why they were not considered for comparison.

• Heritability estimate: seems the authors used LDSC to calculation SNP heritability from GWAS. LDSC is a convenience tool but know to underestimate the trait heritability. It may also cause the discrepancy of heritability of the same traits from different GWAS (e.g., WC from GIANT vs. UKB.

**Have all data underlying the figures and results presented in the manuscript been provided?**

Reviewer #1: Yes

Reviewer #2: None

Reviewer #3: Yes

PLOS authors have the option to publish the peer review history of their article (what does this mean? ). If published, this will include your full peer review and any attached files.

**Do you want your identity to be public for this peer review?** For information about this choice, including consent withdrawal, please see our Privacy Policy .

Reviewer #1: No

Reviewer #2: No

Reviewer #3: **Yes: ** Yan Sun

**Figure resubmission:**
---

## [Editor Report · Decision Letter 1]

9 Jan 2026

Dear Dr Zhao,

We are pleased to inform you that your manuscript entitled "Joint modeling of effect sizes for two correlated traits: characterizing trait properties to enhance polygenic risk prediction" has been editorially accepted for publication in PLOS Genetics. Congratulations!

Yours sincerely,

Chaolong Wang

Academic Editor

PLOS Genetics

Xiaofeng Zhu

Section Editor

PLOS Genetics

Aimée Dudley

Editor-in-Chief

PLOS Genetics

Anne Goriely

Editor-in-Chief

PLOS Genetics

BlueSky: @plos.bsky.social

Comments from the reviewers (if applicable):

Minor Comments

Page12, Line 244-252. & Page 14, Line 271-274 & Page 15, Line 286-289. The interpretation of Tables 1-3 and Tables S1-S3 is currently fragmented across these paragraphs, which distracts from the main narrative. I recommend consolidating the explanations of all tables into a paragraph at the end of the “Real GWAS data” section (e.g., at a location like Page 16, Line 297-302). The redundant text from the aforementioned earlier paragraphs should be removed.

Page 17, Line 338-362. This paragraph largely reiterates the simulation settings that have already been presented in the preceding sections (Page 9, Section “Simulation setting”). To avoid redundancy and improve the manuscript's conciseness, I recommend removing it.

**Data Deposition**

http://datadryad.org/submit?journalID=pgenetics&manu=PGENETICS-D-25-00565R1

**Press Queries**

---

## [Editor Report · Acceptance letter]

PGENETICS-D-25-00565R1

Joint modeling of effect sizes for two correlated traits: characterizing trait properties to enhance polygenic risk prediction

Dear Dr Zhao,

We are pleased to inform you that your manuscript entitled "Joint modeling of effect sizes for two correlated traits: characterizing trait properties to enhance polygenic risk prediction" has been formally accepted for publication in PLOS Genetics! Your manuscript is now with our production department and you will be notified of the publication date in due course.

With kind regards,

Anita Estes

PLOS Genetics

On behalf of:
